# Parents' and guardians' views and experiences of accessing routine childhood vaccinations during the coronavirus (COVID-19) pandemic: A mixed methods study in England

**Sadie Bell** [1‡]*, **Richard Clarke**[2‡], **Pauline Paterson**[3], **Sandra Mounier-Jack**[1]

**1** Faculty of Public Health and Policy, Department of Global Health and Development, London School of Hygiene & Tropical Medicine, London, United Kingdom, **2** Newcastle University Business School, Newcastle University London, London, United Kingdom, **3** Faculty of Epidemiology and Population Health, Department of Infectious Disease Epidemiology, London School of Hygiene & Tropical Medicine, London, United Kingdom

‡ These authors share first authorship on this work.
* sadie.bell@lshtm.ac.uk

**Data Availability Statement:** The full dataset is stored in a data repository: LSHTM Data compass. Data are available to bona fide researchers upon

## Abstract

### Objective

To explore parents' and guardians' views and experiences of accessing National Health Service (NHS) general practices for routine childhood vaccinations during the coronavirus (COVID-19) pandemic in England.

### Design

Mixed methods approach involving an online cross-sectional survey (conducted between 19th April and 11th May 2020) and semi-structured telephone interviews (conducted between 27th April and 27th May 2020).

### Participants

1252 parents and guardians (aged 16+ years) who reported living in England with a child aged 18 months or under completed the survey. Nineteen survey respondents took part in follow-up interviews.

### Results

The majority of survey respondents (85.7%) considered it important for their children to receive routine vaccinations on schedule during the COVID-19 pandemic; however, several barriers to vaccination were identified. These included a lack of clarity around whether vaccination services were operating as usual, particularly amongst respondents from lower income households and those self-reporting as Black, Asian, Chinese, Mixed or Other ethnicity; difficulties in organising vaccination appointments; and fears around contracting COVID-19 while attending general practice.

request and agreement by the study team. Data cannot be shared without restriction as study participants agreed that their anonymised survey and interview data could be accessed by other researchers only. This level of data access was approved by the LSTHM Observational Research Ethics committee (study reference: 21879). Further information on the data and access conditions can be found through the LSHTM Data Compass at: https://doi.org/10.17037/DATA.00001861. Researchers interested in accessing the data are advised to request access through the LSHTM Data Compass page listed above, email the corresponding author or email researchdatamanagement@lshtm.ac.uk.

**Funding:** The research was funded by the National Institute for Health Research (NIHR) Health Protection Research Unit in Immunisation at the London School of Hygiene & Tropical Medicine in partnership with Public Health England. The views expressed are those of the author(s) and not necessarily those of the NHS, the NIHR, the Department of Health or Public Health England.

**Competing interests:** The authors have declared that no competing interests exist.

Concerns about catching COVID-19 while accessing general practice were weighed against concerns about children acquiring a vaccine-preventable disease if they did not receive scheduled routine childhood vaccinations. Many parents and guardians felt their child's risk of acquiring a vaccine-preventable disease was low as the implementation of stringent physical distancing measures (from March 23rd 2020) meant they were not mixing with others.

## Conclusion

To promote routine childhood vaccination uptake during the current COVID-19 outbreak, further waves of COVID-19 infection, and future pandemics, prompt and sustained national and general practice level communication is needed to raise awareness of vaccination service continuation and the importance of timely vaccination, and invitation-reminder systems for vaccination need to be maintained. To allay concerns about the safety of accessing general practice, practices should communicate the measures being implemented to prevent COVID-19 transmission.

## 1 Background

Maintaining the delivery and uptake of routine childhood immunisations is imperative during the coronavirus (COVID-19) pandemic to avoid outbreaks of vaccine-preventable diseases [1–4]. Emerging evidence indicates that the pandemic has caused disruption to the delivery of immunisation programmes globally [5–10]. This disruption is explained by factors including challenges in keeping services running (e.g. due to healthcare worker redeployment or insufficient protective equipment), public fears around accessing healthcare services safely, and movement restrictions [9].

In England, stringent restrictions on physical movement were introduced on 23rd March 2020 to slow the spread of COVID-19, with the general public directed to only leave their homes to: shop for basic essentials, take one form of exercise per day, access medical care or help a vulnerable person, and to travel to and from work if absolutely necessary (where unable to work from home) [11]. People most at-risk from COVID-19 were asked to protect themselves by shielding: staying at home at all times, for at least 12 weeks from 23rd March 2020. In the 3 weeks after the physical movement restrictions were introduced, measles, mumps and rubella (MMR) vaccination counts in children were 19.8% (95% CI: −20.7% to −18.9%) lower than the same period in 2019, before showing signs of improvement in mid-April [7].

Our study explored parents' and guardians' views and experiences of using NHS general practice services for routine childhood vaccination during the early phase of the COVID-19 pandemic. Using the COM-B model [12] we sought to identify factors affecting routine childhood vaccination behaviours during the COVID-19 pandemic in England. We aimed to provide recommendations to inform the way that childhood vaccinations are communicated and delivered during the COVID-19 pandemic, to help improve and maintain routine childhood vaccination uptake.

## 2 Methods

### 2.1 Theoretical framework

We used the COM-B model [12], adapted to vaccination by Habersaat & Jackson [13], to inform the design of study tools and provide a framework for data analysis. The model posits

that capability (C), opportunity (O) and motivation (M) are needed to perform a behaviour (B), such as getting vaccinated. The components of the COM-B model are defined as [12]:

- Capability: *'the individual's psychological and physical capacity to engage in the activity concerned. It includes having the necessary knowledge and skills.'*

- Opportunity: *'all the factors that lie outside the individual that make the behaviour possible or prompt it.'*

- Motivation: *'brain processes that energize and direct behaviour, not just goals and conscious decision-making. It includes habitual processes, emotional responding, as well as analytical decision-making.'*

## 2.2 Design

We used a mixed methods approach involving a cross-sectional online survey, which included fixed and free-text questions, and semi-structured interviews.

**2.2.1 Cross-sectional online survey.** *Recruitment.* We performed an online survey of parents and guardians aged 16 years or older living in England, with a child (or children) aged 18 months or under. Our study specifically focused on parents and guardians with children aged 18 months or under as most childhood vaccinations delivered in general practice in the UK are due before a child reaches 18 months of age (i.e. vaccinations are scheduled at 8 weeks, 12 weeks, 16 weeks and 12 months). There is then a gap in when routine childhood vaccinations are due until a child reaches the age of 3 years and 4 months. Including children up to 18 months meant we could capture the views and experiences of parents and guardians whose children may have been overdue their 12 month vaccinations. Survey recruitment took place between the 19th April 2020 and 11th May 2020. We utilised an online social media recruitment strategy in which our survey was disseminated via Twitter, Facebook, and by email to 284 baby and toddler groups in England. In our recruitment we specially sought to achieve an ethnically representative sample by approaching minority ethnic community groups to advertise the study.

In addition to this Facebook's paid promotion feature was used to target the survey at eligible potential respondents, increasing the reach of the post. The paid promotion feature was used from 22nd April to 6th May 2020 and cost £41. Our Facebook post reached 19,419 people, achieved 3,478 engagements, and was shared 377 times.

*Survey measures.* The survey consisted of four main sections: 1) demographics, 2) likelihood to accept, keep, and/or initiate a vaccination appointment during the COVID-19 pandemic, 3) knowledge and beliefs regarding routine childhood vaccination during the COVID-19 pandemic, and 4) experiences of accessing routine childhood vaccinations during the COVID-19 pandemic.

Demographic questions concerning age, gender, household income, location, employment, marital status, and number and age of children were included. Respondents were asked a series of questions regarding the routine vaccination of their child. These included when their child was due their next vaccination, if an appointment had been scheduled and if they were likely to attend such an appointment. Knowledge about the availability of routine vaccinations was captured in two questions asking if respondents were aware of the government recommendation for routine childhood vaccination services to be maintained, and how certain respondents were that their children could still receive their routine vaccinations during the pandemic.

To measure beliefs and experiences surrounding routine vaccination during the COVID-19 pandemic, respondents were asked if they agreed with statements regarding the importance

and safety of vaccinating their child on schedule, and the difficulty of taking their child for their routine vaccinations. For each statement, respondents indicated their level of agreement on a 5-point Likert scale between *Strongly disagree* to *Strongly agree*.

A statement regarding whether respondents believed that their friends and family felt that they should take their child for a routine vaccination aimed to capture the social norms around routine vaccination during the COVID-19 pandemic.

Finally, respondents were asked about their experiences of accessing routine childhood vaccinations during the COVID-19 pandemic, if they had recently attended, or tried to attend, a routine vaccination appointment. We asked respondents to report any challenges or problems they had experienced in taking their child for vaccinations, or trying to set up an appointment to get their child vaccinated, as a free text response.

The full survey can be found in the supplemental materials.

*Missing data*. In total our recruitment strategy led to 1577 link click throughs. Only those individuals that fully completed the survey were included in the subsequent analysis. This led to the rejection of incomplete data from 316 respondents. A further 9 respondents were rejected due to the reporting of a child, or children, aged over 18 months.

*Analysis*. We performed a forward stepwise logistic regression, using SPSS v.24, to determine factors associated with respondents' lack of awareness of routine vaccination during the COVID-19 pandemic. Age, household income, ethnicity, location, employment, number of children, age of youngest child and date at which the survey was completed were tested for associations. Date was an important additional variable that we decided to include in this model after data collection. This addition was made due to Public Health England (PHE) making an announcement, which subsequently received media attention [2,14], to encourage parents to take their children for routine vaccination during the COVID-19 pandemic. This announcement took place on May 2nd 2020, approximately half-way through our data collection. This date was, therefore, used to segment the sample into those respondents who completed our survey before the announcement and those who completed it afterwards. Respondents did not significantly differ on any demographic variable when compared across this segmentation.

Descriptive statistics and a *t*-test are also reported. Free text responses were analysed thematically in Microsoft Excel. Coding schemes were produced based on the content of the free-text comments.

**2.2.2 Semi-structured interviews.**    *Recruitment and data collection*. On survey completion, respondents were asked to provide their contact details if interested in taking part in a follow-up semi-structured interview. Respondents who had left their details were purposively contacted based on a range of characteristics, including ethnicity, household income, and geographical location. We also purposefully aimed to interview survey respondents who did not provide detailed free-text responses and respondents whose children were overdue a vaccination, or due a vaccination within 4 weeks of taking part in the survey.

Written informed consent was obtained from each participant. Depending on the preference of the participant, the consent form was sent and returned via email or post. Interviews lasted approximately 30 minutes and were conducted via phone. Topic guides, shaped around the content of the questionnaire, were used to support the interviews. Interview participants received a £10 gift voucher as a thank you for their time and contribution. Interviews were conducted between the 27th April and 27th May 2020 by SB and PP, qualitative researchers who have extensive experience in conducting interviews on the topic of vaccination with parents.

*Analysis*. Interviews were transcribed verbatim and analysed thematically using the stages outlined by Braun and Clarke [15]: data familiarisation, coding and theme identification and

refinement. To enhance the rigour of the analysis, coding approaches and data interpretations were discussed between SB, RC, PP and SM-J.

Interviews were coded in NVivo 12 using initial codes generated from the interview topic guide and components of the COM-B model [12].

**2.2.3 Public involvement.** We gained feedback from parents with young children to help refine the survey questions and layout. This aimed to increase the user-friendliness and appropriateness of the survey.

**2.2.4 Ethical approval.** Ethical approval was granted by the London School of Hygiene & Tropical Medicine Observational Research Ethics Committee (study reference: 21879).

## 3 Sample

1252 respondents completed the survey (see survey respondent characteristics in Table 1). Most respondents were female (95.0%; n = 1190), raising a child/children with a partner

**Table 1. Characteristics of survey respondents.**

| Characteristic | | Number of respondents (%) |
|---|---|---|
| **Location** | | |
| | South East | 286 (22.8) |
| | Greater London | 164 (13.1) |
| | North West | 90 (7.2) |
| | East of England | 231 (18.5) |
| | West Midlands | 98 (7.8) |
| | South West | 139 (11.1) |
| | Yorkshire and the Humber | 116 (9.3) |
| | East Midlands | 70 (5.6) |
| | North East | 53 (4.2) |
| | Prefer not to answer | 5 (0.4) |
| **Ethnicity** | | |
| | White:- British | 1082 (86.4) |
| | White:- Irish | 20 (1.6) |
| | White:- Other white background | 76 (6.1) |
| | Black or Black British:- African | 3 (0.2) |
| | Black or Black British:- Caribbean | 1 (0.1) |
| | Mixed:- White and Black Caribbean | 7 (0.6) |
| | Mixed:- White and Black African | 1 (0.1) |
| | Mixed:- White and Asian | 9 (0.7) |
| | Mixed:- Other mixed background | 7 (0.6) |
| | Asian or Asian British:- Indian | 15 (1.2) |
| | Asian or Asian British:- Pakistani | 10 (0.8) |
| | Asian or Asian British:- Bangladeshi | 3 (0.2) |
| | Asian or Asian British:- Other Asian background | 3 (0.2) |
| | Chinese | 2 (0.2) |
| | Other ethnic group not represented by these options | 7 (0.6) |
| | Do not wish to say | 6 (0.5) |
| **Employment status** | | |
| | Working full-time (over 30 hours per week) or on parental leave from full-time employment | 679 (54.2) |

(*Continued*)

**Table 1.** (Continued)

| Characteristic | | Number of respondents (%) |
|---|---|---|
| | Working part-time (less than 30 hours per week) or on parental leave from part-time employment | 404 (32.3) |
| | Homemaker | 114 (9.1) |
| | Student | 12 (0.9) |
| | Unemployed | 13 (1.0) |
| | Other | 23 (1.8) |
| | Prefer not to answer | 7 (0.6) |
| **Household income (GBP)** | | |
| | Under £34,999 | 255 (20.4) |
| | £35,000 - £84,999 | 638 (51.0) |
| | £85,000 and over | 267 (21.3) |
| | Prefer not to answer | 92 (7.3) |
| **Number of children** | | |
| | 1 | 558 (44.6) |
| | 2 | 504 (40.3) |
| | 3 | 148 (11.8) |
| | 4 or more | 42 (3.4) |
| **Age of youngest child** | | |
| | < 2 months | 223 (18.0) |
| | 3–5 months | 330 (26.6) |
| | 6–8 months | 179 (14.4) |
| | 9–11 months | 154 (12.3) |
| | 12–14 months | 218 (17.6) |
| | > 15 months | 138 (11.1) |
| **When is your youngest child next due a vaccination?** | | |
| | They are overdue their next vaccination | 62 (5.0) |
| | In the next 12 weeks | 587 (46.9) |
| | In more than 12 weeks | 565 (45.1) |
| | Do not know | 38 (3.0) |

(97.0%; n = 1214), and identified as being White British, White Irish or White Other (94.1%). The age range of respondents was 18–48 years (Mean = 32.95, SD = 4.565). Median household income was reported as £55,000-£64,999.

Just over half of respondents' (51.8%, n = 649) had a child due for a vaccination within 12 weeks. Of the respondents with a child due a vaccination, 44.8% (n = 291) had a vaccination appointment booked.

43.3% of survey respondents (n = 530) provided details to be contacted for a follow-up interview. In total, 61 parents were contacted to participate. Of these 39 did not respond to recruitment emails, 2 responded initially but did not follow through with an interview, and 19 took part in interviews (18 women and one man). The characteristics of interview participants, and vaccination status of their youngest child, are outlined in Table 2. Each interview participant reported that prior to the COVID-19 pandemic their child/children had received all recommended vaccinations according to the UK schedule.

**Table 2. Characteristics of interview participants and vaccination status of youngest child.**

| No. | Age (years) | Ethnicity | Region | Household Income | Age of youngest child at time of interview | Vaccination status of youngest child |
|---|---|---|---|---|---|---|
| #1 | 25 | Mixed:- White and Asian | South East | Under £34,999 | 13 weeks | Received 8-week and 12-week vaccinations during lockdown |
| #2 | 33 | White:—Other white background | South West | Prefer not to answer | ~ 13 months | Overdue 12 month vaccinations (Practice delayed) |
| #3 | 28 | White British | East Midlands | £35,000 - £84,999 | ~17 weeks | Received 8-week vaccinations. Overdue 12-week vaccinations (Parent delayed–decision supported by family/friends) |
| #4 | 30 | White British | South West | £85,000 and over | 8.5 months | Not due any vaccinations in near future. Child was not due vaccinations during lockdown |
| #5 | 31 | White British | Yorkshire and Humber | £35,000 - £84,999 | 14 weeks | Had 8-week and 12-week vaccinations during lockdown but later than scheduled (10 and 14 weeks) |
| #6 | 39 | White British | South West | £35,000 - £84,999 | 12 weeks | Had 8-week vaccinations during lockdown but later than scheduled (9.5 weeks) (Practice delayed) |
| #7 | 36 | White British | Greater London | £85,000 and over | 3 months | Had 8-week and 12-week vaccinations during lockdown (Practice delayed–due to staff shortages) |
| #8 | 36 | White:—Other white background | Greater London | £85,000 and over | 15 weeks | Had 8-week and 12-week vaccinations during lockdown |
| #9 | 33 | White British | South East | Under £34,999 | 14 months | Overdue 12 month vaccinations (Not contacted by practice but participant wanting to delay) |
| #10 | 34 | Mixed:- White and Black Caribbean | West Midlands | £85,000 and over | 9 weeks | Had 8-week vaccinations during lockdown (Slightly delayed due to difficulties booking appointment) |
| #11 | 34 | White British | South West | £35,000 - £84,999 | 13.5 months | Had 12 month vaccinations during lockdown (Delayed due to practice–practice policy to vaccinate at 13 months) |
| #12 | 39 | White British | East Midlands | £35,000 - £84,999 | 13.5 weeks | Had 8-week and 12-week vaccinations during lockdown (Slightly delayed due to process of registering child at GP, practice, and parent developed potential Covid-19 symptoms) |
| #13 | 33 | Asian or Asian British:—Pakistani | Greater London | £35,000 - £84,999 | 6 weeks | Appointment booked for 8-week vaccinations |
| #14 | 38 | White British | South East | Under £34,999 | 5 months | Had 8-week vaccinations in February (before lockdown), 12-week vaccinations in 1st week. of lockdown and 16-week vaccinations later in lockdown (Slight delay, parent unsure what was going on with vaccinations–waiting for a letter) |
| #15 | 32 | White:—Other white background | East of England | Under £34,999 | 12 months | Had 12 month vaccinations during lockdown |
| #16 | 42 | White British | East of England | £35,000 - £84,999 | 13.5 months | Had 12 month vaccinations during lockdown (day before interview) (Practice delays) |
| #17 | 32 | White British | South West | Under £34,999 | 12 months 3 weeks | Had 12 month vaccinations during lockdown (Slight delay as asked by practice to book appointment later) |
| #18 | 34 | Chinese | Greater London | £85,000 and over | 11 weeks | Had 8-week vaccinations during lockdown (Delays due to difficulties booking appointment) |
| #19 | 25 | Mixed:- White and Black Caribbean | West Midlands | £35,000 - £84,999 | 12 weeks | Had 8-week and 12-week vaccinations during lockdown |

## 4 Findings

The following sections present data according to the COM-B model, identifying factors affecting routine childhood vaccination behaviour during the COVID-19 pandemic by drawing on fixed-response and free-text components of the survey, and the qualitative interviews. Quotes from interviews and free-text responses are provided in Table 3 to illustrate our findings.

Table 4 highlights free-text responses given when parents and guardians were asked to provide details on any issues, challenges, or problems they experienced in taking their child for vaccinations or trying to set up a vaccination appointment. Of the 670 survey respondents who had tried to obtain routine childhood vaccinations since 23rd March, almost a quarter

**Table 3. Interview and free-text responses illustrating components of the COM-B model.**

| Component of COM-B model | Description | Illustrative quotations from interviews or free-text response |
|---|---|---|
| **Capability**<br>*'the individual's psychological and physical capacity to engage in the activity concerned. It includes having the necessary knowledge and skills.'* | Awareness of vaccination service continuation | *'the only reason I really knew that they were going ahead is because my friend's got a baby that's three weeks older and she'd had hers, so I knew that they were going ahead. But otherwise, yeah, I wouldn't have been sure at all.' (Interviewee #10)* |
| | Unable to attend practice due to illness/COVID-19 symptoms | *'I had got a temperature... then we were isolating for two weeks and that meant that I had to phone up and change the appointment.' (Interviewee #12)*<br>*'Delayed due to coronavirus symptoms within the household (turned out not to be coronavirus but had to isolate anyway).' (Survey ID#51)* |
| | Knowing what to expect at vaccination appointments | *'...if you knew in advance what had been done in the surgery and how the rooms were set out and things like that, that would sort of make you feel a bit more comfortable about it' (Interviewee #12)* |
| **Motivation**<br>*'brain processes that energize and direct behaviour, not just goals and conscious decision-making. It includes habitual processes, emotional responding, as well as analytical decision-making.'* | Safety concerns and how these were allayed by positive experiences of attending GP practice (i.e. personal experiences and those of other parents and guardians) | *'I want her to get her vaccinations but then it's getting there if it's a rainy day... I don't want to get on a bus because it's been quite busy here anyway, which is not ideal and albeit, yes infection rates are low but I don't want to create an unnecessary risk.' (Interviewee #13)*<br>*'I was a bit nervous about going to the GP... in the end I called, um, I rang my friend, she had a newborn as well, and she explained to me what happened when she went to the GP and she got given a mask and gloves and she felt quite safe in her appointment. So I thought OK, it's better to get him vaccinated because there's a risk of other diseases as well. I felt a lot more secure.' (Interviewee #18)*<br>*'So the eight weeks before it was kind of lockdown, I was really anxious because it was all up in the air, not really sure what you're doing, and [I] went to the waiting room and it was really busy and it was a really small waiting room as well... So that was quite anxious, but then the lockdown, when the lockdown started then I was really anxious... But then it was a completely different experience because they had shut the doctor's surgery, so no one could just walk in. Pressed the buzzer, the receptionist asked if I had a cough or a temperature before she'd let me in. Which she did, and then when I went in the doctor's there was no one there, and there was no one in the waiting room because they had blocked it so there was 10 minutes between each appointment.' (Interviewee #14)* |
| | Perceived importance of vaccinating children and weighing up risk and benefits of getting children vaccinated during lockdown | *'In the news, I think at the time, there's been a lot about measles outbreaks, so they [most people] were saying that they didn't want to put their child at risk of other things that they were being vaccinated against just because of the pandemic, they don't want those children to fall behind in the schedules. I know one of the things that worried me was if a lot of people were delaying it, that once the lockdown was over, there might be a backlog and you not be able to get an appointment to get your child vaccinated.' (Interviewee #19)*<br>*'At the moment, you know, we're not going anywhere that she could pick up anything including measles or mumps or whatever. So, I'm okay with her holding off until we're going to be out and about but when we are going to out and about again because obviously, we can't stay inside forever, it will have to be a priority. Because the last thing we need is for her to pick up something else and [laughs]. But it's just, it's, we're trying to sort of balance it, you know, we don't want to go anywhere to risk her getting anything at the minute and the GP's surgery just feels like—I could be wrong, but it just feels like the sort of place that you really want to avoid.' (Interviewee #9)* |
| **Opportunity**<br>*'all the factors that lie outside the individual that make the behaviour possible or prompt it.'* | **Physical opportunity** | |
| | Challenge registering child at general practice | *'If anything, that [registering child at practice] was quite a stressful thing and I think that if I was somebody maybe who was finding parenting in general harder at that point in time or had a lot more on in life, that would, you know, it potentially could've resulted in either, you know, not getting the jabs in a timely manner... If I wasn't as passionate about, yeah, about the fact that I want these vaccinations to happen. Yeah, would I have let it go? And well, I'll sort it out in a few weeks.' (Interviewee #12)* |
| | Difficulty booking appointments | *'when I made the appointment on Friday almost like a couple of minutes later I got a text saying your appointment is now cancelled, it's now a phone call appointment, because I think in their system they automatically just send it to everyone that's got an appointment saying it's cancelled... I called them straight away saying how are you going to do the jabs, you know, and she explained oh, sorry, it's just automatic. But I think for other parents, if they got that message they'd think it's just cancelled.' (Interviewee #1)*<br>*'I know of a number of parents who are choosing to delay vaccinations. Ideally GPs should book in vaccinations (rather than wait for patients to do it) and emphasise the importance.' (Survey ID 979)* |
| | 6–8 week checks missed | *'The only issue we have faced is that her 6-week check was cancelled by our GP practice due to covid-19 restrictions yet she wasn't allowed her first set of immunisations until she'd had the check. There seemed to be no guidance on how the surgery should handle this. In the end, I had to 'refuse' some of the checks (the docs couldn't perform them due to the restrictions) just so my daughter could have her immunisations.' (Survey ID 187)* |
| | Managing childcare | *'I think if I felt that you know, either [husband] could either take him or if we could go with both the kids and there'd be that flexibility for us to have support if needed then we would book ahead and do it' (Interviewee #3)*<br>*'Had to find childcare for the siblings as husband is a keyworker at work and you can't take siblings into the GP. No one else can look after them as that breaks the social distancing rules set by the government.' (Survey ID 177)* |
| | Availability of information on what to expect after vaccinations | *'[I would like] maybe an information sheet or something from the nurses to say if there was anything, yeah, I don't know. Well, I think we all know the symptoms of Covid and what to look out for, but it's difficult because, you know, there's like how do you determine the difference? I know it's very rare but there's that Kawasaki syndrome is it that can be a, I don't know. A friend of mine actually like shared the symptom list and sort of pictures on Facebook earlier and I was like maybe if they'd had something like that. Because if you've got a fever you might be having some reactions because the vaccine can give some funny reactions and they can get a rash and things like that as a reaction to the vaccine, but it's like how do you tell the difference between a reaction to the vaccine and, you know, potentially something different, something more sinister?' (Interviewee #11)* |
| | **Social opportunity (social influences) (e.g. social cues, norms)** | |
| | Social norms | *'They [family members] seemed to think that it was natural that they should go ahead. My friends were a bit less and were being told not to go to the grocery store if you don't have to. Does it really hurt if we're not going outside anyway? The baby's not going to get measles if we're stuck in the house.' (Interviewee #8)* |

**Table 4. Free-text responses given when parents and guardians were asked to provide details on any issues, challenges, or problems they experienced in taking their child for vaccinations or trying to set up a vaccination appointment.**

| | | | |
|---|---|---|---|
| Capability | Appointment cancelled because of illness/possible COVID-infection affecting member of household | 8 | 5.3% |
| Motivation | Felt delivery of vaccination at practice itself was unsafe/problematic | 9 | 5.9% |
| | Felt unsafe to vaccinate child/children | 2 | 1.3% |
| Opportunity | Problem with appointment bookingIssues reported:<br>• GP practice releasing appointments week-by-week. Parents unable to book appointments in advance. No appointments available (n = 47)<br>• Only online appointments available (n = 20)<br>• Confusion amongst healthcare workers about whether routine childhood vaccinations going ahead e.g. GP unsure about process, health visitor saying all appointments postponed (n = 8)<br>• GP practice advised parents to wait and attend at a later date for vaccinations (n = 2)<br>• Other difficulties in booking appointments, including GP practice asking for an appointment letter to arrive before parents could book (n = 4) | 81 | 53.3% |
| | Appointment cancelled | 43 | 28.3% |
| | Appointment cancelled but rebooked successfully | 11 | 7.2% |
| | Appointment offered, or delivered, in another GP practice | 10 | 6.6% |
| | GP practice closed | 6 | 3.9% |
| | Issue with registration of child at GP practice (birth certificate unavailable) | 5 | 3.3% |
| | Only some vaccinations available | 5 | 3.3% |
| | Only able to attend with one parent/guardian | 5 | 3.3% |
| | No vaccinations offered | 4 | 2.6% |
| | Appointment cancelled due to healthcare worker sickness | 3 | 2.0% |
| | Issues with accessing the BCG vaccine | 2 | 1.3% |
| | Stock issues–out of vaccines | 1 | 0.7% |

(23.9%, n = 160) reported difficulties in organising or accessing vaccination appointments and provided a free-text reason for this (see Table 4). Table 4 is drawn upon in the following sections.

## 4.1 Capability

**Awareness of vaccination service continuation.** Although most survey respondents (74.4%; n = 931) had heard the national recommendation that routine childhood vaccinations should go ahead as normal during the COVID-19 pandemic, one in four respondents were not aware of this recommendation (25.6%; n = 321). One in five said that they were *very uncertain* (5.5%; n = 69) or *somewhat uncertain* (16.9%; n = 209) about whether their child could still receive vaccinations during lockdown.

Several interview participants said they had been unsure about whether routine childhood vaccinations were being classed as an 'essential service' and operating as usual during the COVID-19 pandemic, particularly at the beginning of lockdown. Interview participants generally reported that their knowledge about the continuation of routine vaccinations had come through communication with other parents and guardians, often through friends, parenting forums, social media networks, and Mobile Apps for parents. Several participants reported that they could not find any information on the NHS website about vaccinations continuing as usual. Some participants cited the news as a source of information on vaccinations, while others reported avoiding the news, which they said provoked anxiety.

Fewer participants had heard about vaccines continuing through receipt of a letter inviting them to book a vaccination appointment, or by contacting their GP. Several interview participants held back from phoning their general practice to find out if vaccinations were going ahead, as they felt practices were '*busy enough*' and did not want to add to their workload.

Interview participants felt that more information about vaccinations going ahead should have been provided, with several suggesting parents receive a text-message or call from their general practice or health visitor, and that up to date information be added to the NHS and general practice websites. Others recommended stronger and more prompt national-level communication messages, from the government, the NHS and public health bodies.

**Factors associated with a lack of awareness of vaccination service continuation.** A total of 1117 responses were included in the logistic regression. Four variables were independently associated with a lack of awareness of routine vaccine during the covid-19 lockdown (household income, age of youngest child, ethnicity and date at which survey was completed), all of which were also significant in the final model (omnibus chi-square = 61.423, df = 9, $p < .001$). The Hosmer and Lemeshow test demonstrates that the model adequately fits the data chi-square = 8.677, df = 7, $p = .277$. Table 5 gives coefficients and the Wald statistic and associated degrees of freedom and probability values for each of the predictor variables.

The findings of this analysis indicate that respondents from ethnic minority backgrounds (i.e. Black, Asian, Chinese, Mixed or Other Ethnicity) were almost three times more likely to be unaware of the recommendation of vaccination service continuation compared to White ethnic groups (i.e. White British, White Irish, White Other) (95%CI: 1.73–5.32). Similarly, respondents reporting a household income of <£35,000 per annum were 1.5 times more likely to be unaware of the recommendation compared to respondents with an annual household income of £35,000 - £84,999 (95%CI: 1.08–2.12). Respondents were more likely to be aware of the recommendation after the May 2$^{nd}$ announcement from PHE than before (70.3% aware

**Table 5. Logistic regression of lack of awareness of routine childhood vaccination service continuation during lockdown.**

| Variable | | Univariate analysis | | | Multivariate analysis | | |
|---|---|---|---|---|---|---|---|
| | | Sig (p) | OR | 95% CI | Sig (p) | OR | 95% CI |
| **Income** | | **.034** | | | **.049** | | |
| | Low Income <£35,000 (n = 240) | .009 | 1.539 | 1.112–2.131 | .016 | 1.512 | 1.079–2.120 |
| | Medium Income £35,000 - £84,999 (n = 617)† | - | - | - | - | - | - |
| | High Income >£85,000 (n = 260) | .422 | 1.147 | .821–1.603 | .807 | 1.044 | .738–1.477 |
| **Age of child** | | **< .001** | | | **< .001** | - | - |
| | <2 months (n = 199) | .053 | .635 | .400–1.006 | .040 | .595 | .362 –.977 |
| | 3–5 months (n = 299) | .025 | .616 | .403 –.940 | .055 | .646 | .413–1.010 |
| | 6–8 months (n = 166)† | - | - | - | - | - | - |
| | 9–11 months (n = 139) | .098 | 1.475 | .930–2.338 | .122 | 1.475 | .902–2.414 |
| | 12–14 months (n = 192) | .089 | .671 | .424–1.063 | .094 | .652 | .398–1.076 |
| | >15 months (n = 122) | .084 | 1.518 | .945–2.436 | .046 | 1.669 | 1.009–2.759 |
| **Ethnicity** | | **< .001** | | | **< .001** | - | - |
| | White † (n = 1064) | - | - | - | - | - | - |
| | BAME (n = 53) | < .001 | 2.785 | 1.699–4.567 | < .001 | 3.029 | 1.726–5.315 |
| **PHE announcement date cut** | | **< .001** | | | **< .001** | - | - |
| | Before (n = 588) | - | - | - | - | - | - |
| | After (n = 529) | < .001 | .631 | .487 - .817 | .001 | .625 | .471 - .829 |

† Reference category.

before vs. 79.0% aware after), and there is some indication that parents of children less than 2 months of age were more likely to be aware of the recommendation compared to children between 6–8 months of age.

**Unable to attend vaccination appointments due to illness/COVID-19 symptoms.** A minority of respondents had experienced difficulties in attending vaccination appointments due to a member of the household becoming unwell, possibly with COVID-19 infection (5.3%, n = 8; see Table 4). One interview participant had developed symptoms of COVID-19 and consequently needed to isolate for 2 weeks, resulting in delays to her daughter's vaccination.

**Knowing what to expect at vaccination appointments.** Many interview participants had felt nervous about taking their child for vaccinations before the COVID-19 pandemic (i.e. they were worried about vaccine side effects, and concerned about their child being upset immediately after injection). The COVID-19 pandemic had generated additional fears for parents around the safety of attending vaccination appointments and the risk of catching COVID-19. Several participants discussed their anxiety at not knowing ahead of their appointment what measures had been put in place to keep patients safe. First-time parents, taking their child for their first set of vaccines, appeared particularly nervous about their appointments as they had no benchmark of what to expect. Participants also felt that more information about new measures in place to ensure safety for all (i.e. social distancing measures, protective equipment wearing, and increased times between appointments to reduce the flow of patients) should have been given to reassure parents when booking their appointment or on the general practice website.

## 4.2 Motivation

**Safety of vaccinating children during the pandemic.** Most survey respondents (72.7%, n = 911) strongly or somewhat agreed with the statement '*During the coronavirus (COVID-19) pandemic, I feel it is safe to go to the general practice to vaccinate my child/children on time for their routine vaccinations*'. One in five respondents (20.2%, n = 253) disagreed to some extent with the statement (Fig 1).

Several participants reported that concerns around the safety of accessing general practice had led to delays in them chasing up vaccination appointments, or to waiting to attend after the peak of infections had passed.

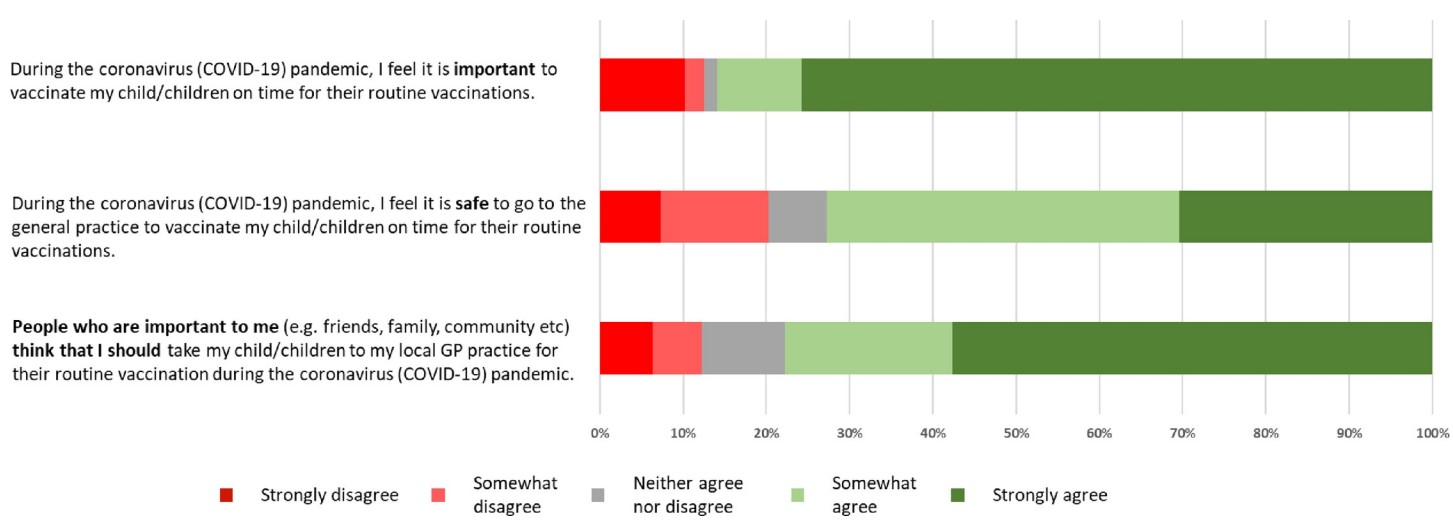

**Fig 1. Parents and guardian's beliefs regarding routine childhood vaccination during the COVID-19 pandemic in England.**

The majority of interview participants discussed having a positive experience once they had attended a vaccination appointment, reporting that they were reassured by the safety measures taken to prevent the spread of COVID-19. These included patients being screened for COVID-19 symptoms before attending, waiting outside the practice and using an intercom system to be admitted, screens between patients and receptionists, the provision of hand sanitiser and masks, waiting rooms being kept quiet by leaving longer time periods between appointments, doors being opened by healthcare professionals (to avoid patients needing to touch door handles), staff wearing protective equipment, and social distancing being maintained. Having a positive experience motivated parents to reassure and encourage others to take their children for vaccinations, and also reassured participants about attending subsequent appointments.

One interview participant expressed the challenge of travelling to vaccination appointments when advised not to use public transport.

**Perceived importance of vaccinating children.**   Most respondents strongly (75.7%, n = 948) or somewhat agreed (10.1%, n = 127) with the statement '*During the coronavirus (COVID-19) pandemic*, *I feel it is important to vaccinate my child/children on time for their routine vaccinations*'. While there was less overall disagreement with this statement (12.5%, n = 157) compared to that of safety, the majority of respondents that disagreed selected the strongly disagree option (10.2%, n = 128) (Fig 1).

All interview participants said vaccinating their children was important; however, this was balanced against their concerns over vaccinating their children during the pandemic. Interview participants discussed the weighing up of perceived risks and benefits of taking their children for vaccination. Concerns about contracting COVID-19 while travelling to or accessing general practice were weighed against concerns about their child contracting a vaccine-preventable disease if they did not vaccinate. Many parents also reported that during lockdown they did not feel their child was at risk of acquiring a vaccine-preventable disease, as they were not mixing with other people.

**Likelihood of keeping existing vaccination appointments and booking vaccinations.** Survey respondents indicated that they were still strongly motivated to maintain existing vaccination appointments and to book due vaccinations during the COVID-19 pandemic. For survey respondents with upcoming vaccination appointments already booked, the majority reported that they were very likely (93.8%, n = 273) or likely (4.8%, n = 14) to keep these appointments.

For those who did not already have an appointment booked, and were due vaccinations within 12-weeks of completing the survey, the majority reported that they were very likely (75.3%, n = 220) or likely (16.4%, n = 48) to contact their general practice to organise an appointment.

**Perceived difficulty of making vaccination appointments.**   Just over a quarter of all respondents (26.7%; n = 335) agreed to some extent with the statement '*I feel that the current constraints due to the coronavirus (COVID-19) pandemic would make it difficult for me to make a vaccination appointment at my general practice*'.

Respondents who had taken their child for vaccinations during lockdown were significantly less likely to agree that it would be difficult to obtain a vaccination appointment (Mean = 2.97, SD = 1.188) than those that had not taken their child for vaccinations (Mean = 2.15, SD = 1.325), $t (1250) = 11.483$, $p < .001$ (Fig 2). This indicates that parents' and guardians' experiences of making appointments were more positive than anticipated.

## 4.3 Opportunity—Physical

**Challenge registering child at general practice.**   A minority of respondents (3.3%, n = 5) (Table 4) reported issues registering their newborn babies in general practice, particularly

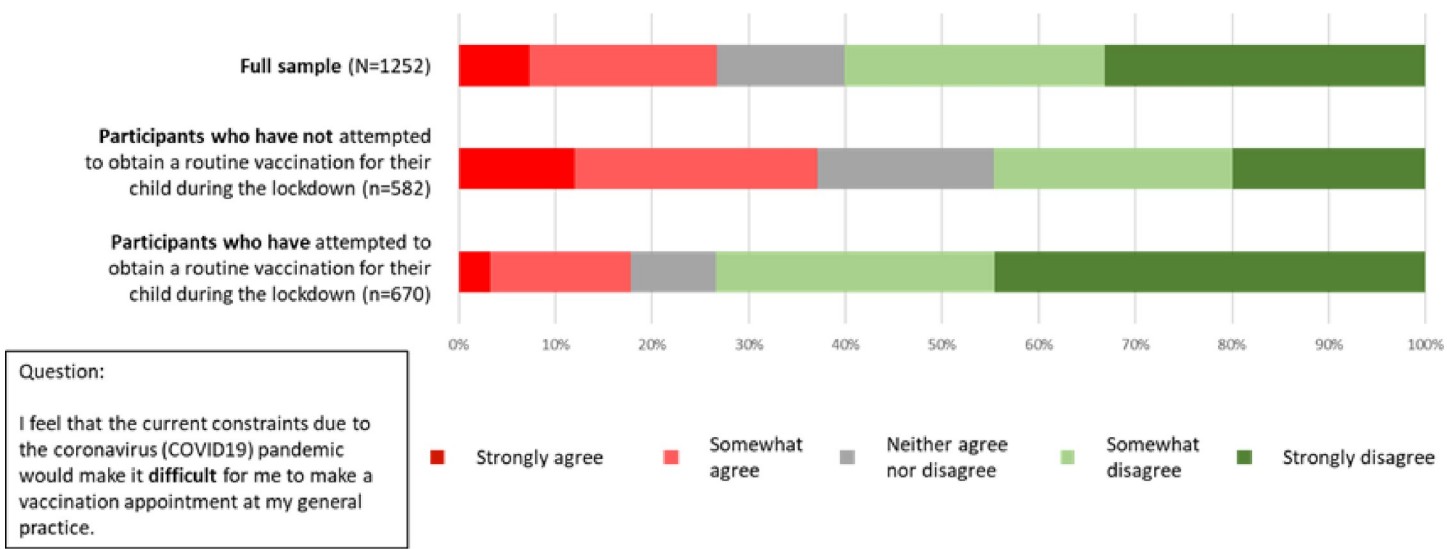

**Fig 2. Parents' and guardian's perceptions of the difficulty of making an appointment for routine childhood vaccination during the coronavirus (COVID-19) pandemic.**

when they had also been unable to obtain a birth certificate. Interviewees discussed the time-consuming process of completing relevant paperwork at their general practice and being unable to register the child remotely. Participants reported a need to 'chase' their GP practice multiple times about paperwork requirements and felt the onus was fully on them to get their child registered.

One interview participant had found it particularly stressful to organise their child's registration, and felt that for some parents and guardians the experience could have led to them not getting their children registered and vaccinated in a timely manner.

**Difficulty booking appointments.** Of the 160 survey respondents reporting difficulties in organising or accessing vaccination appointments that had left a free-text reason for this (Table 4), 53.3% (n = 81) had difficulties in booking appointments.

In many instances, interview participants who contacted their GP practice were told to phone back weeks after their initial contact to book an appointment, or told that appointments could not be booked in advance. During interviews, some parents and guardians reported that GP receptionists were not sure themselves whether routine childhood vaccinations were still going ahead, particularly at the beginning of the lockdown. Other issues mentioned included: parents not being able to book appointments in advance (with appointments only released week-by-week), only online appointments being offered, and appointments being cancelled (see Table 4). Interview participants felt that the onus had been placed on them to push for appointments to be organised and many felt that less proactive parents, or parents less adamant on getting their children vaccinated, may have given up trying to organise an appointment.

Several interview participants also discussed receiving confusing text-messages about their appointment from their GP practice, that stated their appointments had been cancelled and they should not attend the GP practice. These automated messages were received by some participants several weeks into the lockdown and may have led to some parents not attending the vaccination appointment.

A minority of survey respondents (2.6%; n = 4) reported that no vaccines were being offered by their GP practice. Several survey respondents (6.6%, n = 10) had needed to attend another general practice for vaccinations (Table 4).

**6-8-week baby checks and postnatal checks missed.**   Two interview participants discussed that they were unable to access 6–8 week postnatal or baby checks at their GP practice. These parents felt that baby checks were a pre-requisite for vaccination and were frustrated that they had not been performed.

**Managing childcare.**   Survey respondents (3.3%, n = 5) discussed the challenge of only being able to attend the GP practice with one parent and one child (Table 4). One interview participant who had two children and a shielding husband, and whose eldest son had previously reacted to a vaccine and needed to attend hospital, did not feel able to take her youngest child for his vaccines due to worries about how they would manage childcare if he also had an adverse reaction following immunisation.

**Availability of information on what to expect after vaccinations.**   Several interview participants reported that access to information about vaccines, particularly advice on what to do after vaccination (e.g. if the child developed a temperature), was not sufficiently provided during their appointment. Participants understood that practice nurses wanted to complete appointments as quickly as possible to reduce contact time, but this could be to the detriment of fully informing parents. One participant voiced concerns that side effects of vaccination could mimic those of COVID-19 infection and wanted advice on how to distinguish between the two.

## 4.4 Opportunity—Social

**Social norms.**   Survey respondents were asked about their level of agreement with the statement: *'People who are important to me (e.g. friends, family, community etc.) think that I should take my child/children to my local GP practice for their routine vaccination during the coronavirus (COVID-19) pandemic'*. The majority of participants (77.8%; n = 168), for whom the conversation had arisen, reported that the norm amongst social networks was to vaccinate.

Interview participants reported that some family members were shocked and questioned whether children should be vaccinated; however, most were supportive of the decision to vaccinate their child.

## 5 Discussion

### 5.1 Principal findings and implications for policy and practice

This mixed methods study explored parents' and guardians' views and experiences of childhood vaccination during the coronavirus (COVID-19) pandemic in England, at a time when stringent lockdown measures had been implemented and the number of COVID-19 cases was peaking. We used the COM-B model [12,13] to identify factors affecting routine childhood vaccination behaviour, providing insights as to why routine childhood vaccine uptake dipped in England during lockdown [7].

Our findings indicate that parents and guardians in England continued to view vaccines as important during the early phase of the COVID-19 pandemic (March to May 2020), with similar levels of agreement on the importance of vaccinating children to the pre-COVID period [16]. Most parents and guardians wanted to vaccinate their children during the COVID-19 pandemic; however, they experienced barriers that influenced their capability, motivation and opportunity to vaccinate their children.

Parents and guardians reported difficulties in booking vaccination appointments and not receiving vaccination invites and reminders. It is well-documented that invitation-reminder

systems are one of the most effective interventions for improving immunisation rates [17,18]. Our findings indicate that these systems were not fully maintained early in the pandemic (during the first national lockdown), as parents and guardians reported not receiving invites or reminders from their GP practice, and instead more onus was placed on them to remember when vaccinations were due and to organise appointments.

Parents and guardians reported a lack of clear national guidance on whether routine vaccinations were still going ahead as planned, particularly at the start of lockdown. These uncertainties were rooted in the government rhetoric to "*stay at home, protect the NHS and save lives*" and advice to avoid attending GP practices and postpone 'non-essential' appointments [19]. Evidence suggests that the public has been concerned about accessing the NHS due to fears around contracting COVID-19 and not wanting to put pressure on services, with reports of reduced general practice and A&E attendance, particularly early in the lockdown [20–22]. Many parents in our study were unclear about whether vaccinations were classed as essential, and most interview participants had learnt that vaccinations were going ahead through their social networks and parenting groups rather than from their GP practice.

Parents and guardians expressed concerns when thinking about the prospect of safety travelling to and attending general practice for fear of themselves or their child contracting COVID-19. Generally, however, parents and guardians who had attended appointments with their children for vaccinations had their concerns alleviated and felt safe. To allay concerns about attending practice, more information needs to be made available for parents on what to expect when attending—i.e. what safety measures are being taken and how the process has changed to prevent the spread of COVID-19.

One concerning aspect of our quantitative findings was the disparity in awareness of the COVID-19 routine vaccination policy across income and ethnicity groups. With the fact that Black, Asian and minority ethnic and low-income groups have been disproportionately affected by COVID-19 [23] it is perhaps understandable that routine vaccination may take a back seat to other more pressing concerns, however, the long-term consequences of under-immunisation within these groups could cause additional health burdens. While the Public Health England national campaign appeared to be effective in increasing awareness of the policy, additional targeted communications could also be beneficial.

## 5.2 Strengths and limitations

Using a mixed methods approach in this study allowed for a more complete insight into parents' and guardians' views and experiences of vaccinating young children during the COVID-19 pandemic in England than could have been achieved using one method only. As part of the survey, rich and detailed information was obtained in free-text responses and findings from the interviews supplemented this.

A limitation of the interviews was that only 19 of 61 invited respondents took part, which may have been partly due to respondents not seeing the invitation email or not having the time to participate. It may be that those that took part in interviews had a stronger interest in vaccinations and a different view to those that did not take part.

Our recruitment strategy, using social media, achieved a high number of responses. Although geographically representative, our respondents were not overly representative in terms of household income and ethnicity. This may have been reflective of using an online recruitment approach which may have biased who took part in the research. Most of our survey respondents were White (White British, White Irish or White Other) and reported relatively high annual household incomes (median household income £55,000-£64,999). Further to this, an exploration of gender as an influencing factor in the study was not possible due to

the limited number of male parents and guardians completing the survey, and we did not collect data on additional demographic factors that may have been of interest, such as education level, health literacy and religious beliefs.

As inequalities in vaccination access and uptake are found in minority ethnic and lower income groups [24–26], the findings may not capture the views and experiences of people who face the greatest barriers to vaccination.

## 6 Conclusion

Overall, during the early phase of the coronavirus (COVID-19) pandemic, parents and guardians strongly believed in the importance of vaccinating their children and most parents wanted to get their children vaccinated. Despite this, several barriers to uptake were identified, particularly related to awareness of routine vaccinations going ahead, concerns around the safety of attending general practice, difficulties in booking appointments, and not receiving vaccination reminders from their GP practice.

We provide recommendations to inform effective vaccination programme delivery during the current COVID-19 outbreak, further waves of COVID-19 infection, and future pandemics. Our main recommendations are to improve awareness of vaccination service continuation through GP-level and national-level communication streams, to maintain invitation-reminders systems, and to ensure that parents are aware of measures being implemented in general practice to prevent COVID-19 transmission.

## Supporting information

**S1 File.**
(PDF)

## Acknowledgments

We would like to thank the baby and toddler groups who helped share the online survey with potential participants. We are especially grateful for the time and contribution of all parents and guardians who took part in the study.

We would also like to thank Jo Yarwood and David Green at Public Health England for their feedback in designing the study, particularly in developing the survey tool.

## Author Contributions

**Conceptualization:** Sadie Bell, Richard Clarke, Pauline Paterson, Sandra Mounier-Jack.

**Data curation:** Sadie Bell, Richard Clarke.

**Formal analysis:** Sadie Bell, Richard Clarke, Pauline Paterson, Sandra Mounier-Jack.

**Investigation:** Sadie Bell, Richard Clarke, Pauline Paterson, Sandra Mounier-Jack.

**Methodology:** Sadie Bell, Richard Clarke, Pauline Paterson, Sandra Mounier-Jack.

**Validation:** Sadie Bell, Richard Clarke, Pauline Paterson, Sandra Mounier-Jack.

**Writing – original draft:** Sadie Bell, Richard Clarke.

**Writing – review & editing:** Pauline Paterson, Sandra Mounier-Jack.

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
