## [Decision Letter · Decision Letter 0]

3 Oct 2020

PONE-D-20-26785

Parents’ and guardians’ views and experiences of accessing routine childhood vaccinations during the coronavirus (COVID-19) pandemic: A mixed-methods study in England

PLOS ONE

Dear Dr. Bell,

Thank you for submitting your manuscript to PLOS ONE. After careful consideration, we feel that it has merit but does not fully meet PLOS ONE’s publication criteria as it currently stands. Therefore, we invite you to submit a revised version of the manuscript that addresses the points raised during the review process.

We look forward to receiving your revised manuscript.

Kind regards,

Khin Thet Wai, MBBS, MPH, MA (Population & Family Planning Resear

Academic Editor

PLOS ONE

Journal Requirements:

2. Please include additional information regarding the qualitative interview guide used in the study and ensure that you have provided sufficient details that others could replicate the analyses. For instance, if you developed a interview guide as part of this study and it is not under a copyright more restrictive than CC-BY, please include a copy, in both the original language and English, as Supporting Information.

3. In the Methods, please discuss whether and how the interview guide was pre-tested. If this did not occur, please provide the rationale for not doing so.

Reviewers' comments:

Reviewer's Responses to Questions

**Comments to the Author**

1. Is the manuscript technically sound, and do the data support the conclusions?

Reviewer #1: Yes

Reviewer #2: Yes

2. Has the statistical analysis been performed appropriately and rigorously? 

Reviewer #1: Yes

Reviewer #2: Yes

3. Have the authors made all data underlying the findings in their manuscript fully available?

Reviewer #1: No

Reviewer #2: Yes

4. Is the manuscript presented in an intelligible fashion and written in standard English?

Reviewer #1: Yes

Reviewer #2: Yes

5. Review Comments to the Author

Reviewer #1: This is a comprehensive and interesting study about exploring the experiences of routine childhood vaccination in the midst of COVID-19 pandemic. This is an important area of study given the extensive data form diverse settings.

The manuscript is written comprehensively and research design is appropriate with proper integration of quantitative and qualitative findings. Exploring also the aspects of healthcare providers for vaccination during pandemic will provide complete picture of lessons and programmatic implications in improving vaccination services during pandemic situation. Overall, the study provide valuable insights for barriers in uptake immunization services during pandemic situation with high applicability for local and global context. The comments were shown below.

Introduction

•The problem is properly introduced and the justification is robust enough to support the objective of the study.

Methods

•Line 97: Why the study included only 18 months or under children although GP surgery provided immunization up to more than 3 years old?

•Line 143: What about the association between awareness of routine vaccination with the gender and education status of parents or guardians?

•In recruiting and collecting data for semi-structured interview, there is no information about characteristics of interviewers and saturation of data.

•Line 165: Is it online based written consent?

•Line 169 & 174: What are meanings of abbreviations; SB, PP, RC and SM-J?

•Line 185 & 198: Please check appropriate reference for “Error! Reference Source not found”

•Line 194 – 197: The selection criteria for choosing 61 parents among 530 with detail contact information is not clear. Only 19 out of 61 invited (31%) were interviewed which might lead to non-response bias.

•In table 2: There are only 2 respondents whose children were overdue a vaccination. It is questionable whether the data is saturated enough in exploring the barriers of taking vaccination among such group.

Finding

•Table 5: Why medium income level and age group 6-8 months were taken as reference in logistic model?

•Line 237 – 247: The order of factors interpreted in text is better to follow the order as shown in table .

•Line 248 – 264: It would be better to include sub-heading for these paragraphs as these paragraphs provide information not directly linked with factors associated with lack of awareness.

•Line 268: Please check appropriate reference for “Error! Reference Source not found”

•Line 369: Who or which organization sent confusing text-messages?

Discussion and conclusion

•Although the scope of the study is big, the discussion provides solid messages to support the conclusion. In presenting discussion, it would be better to follow the sequence of information; capability, motivation and opportunity as written in results part for better linkage.

Reviewer #2: This is well-organized/planned survey for the effects of pandemic on routine immunization for infants below 18 months. This is very important point for ongoing COVID-19 pandemic (potential further wave) or other pandemics in future.

Authors clearly defined their aims and methods using prompt references.

There are some minor points (maybe limitations)

- Number of interviewed participants is low; only 19. Regarding enrollment number, it is difficult to evaluate this small group response.

- Your title includes terms "parents", however as usual for studies during infancy, 90%of responses came from mothers. This is not parent's view, this is mother's perspective.

- Table 4-5 (colored-component of COMb) is busy to follow-up, maybe move to supplementary file or need to re-organize.

- I am not sure the Journal Policy (page limitations), your Results section is too long and difficult to-follow up.

6. PLOS authors have the option to publish the peer review history of their article (what does this mean?). If published, this will include your full peer review and any attached files.

Reviewer #1: No

Reviewer #2: **Yes: **Ener Cagri Dinleyici

---

## [Author Response · Author response to Decision Letter 0]

18 Nov 2020

Reviewers' comments:

Reviewer's Responses to Questions

Comments to the Author

1. Is the manuscript technically sound, and do the data support the conclusions?

Reviewer #1: Yes

Reviewer #2: Yes

2. Has the statistical analysis been performed appropriately and rigorously? 

Reviewer #1: Yes

Reviewer #2: Yes

3. Have the authors made all data underlying the findings in their manuscript fully available?

Reviewer #1: No

We have added the following to the manuscript (lines 501-504).

‘Data Availability Statement: The full dataset will be made available to bona fide researchers upon request and agreement by the study team. Further information on the data and access conditions can be found through the LSHTM Data Compass at: https://doi.org/10.17037/DATA.00001861.’

Reviewer #2: Yes

4. Is the manuscript presented in an intelligible fashion and written in standard English?

Reviewer #1: Yes

Reviewer #2: Yes

5. Review Comments to the Author

Reviewer #1: This is a comprehensive and interesting study about exploring the experiences of routine childhood vaccination in the midst of COVID-19 pandemic. This is an important area of study given the extensive data form diverse settings.

The manuscript is written comprehensively and research design is appropriate with proper integration of quantitative and qualitative findings. Exploring also the aspects of healthcare providers for vaccination during pandemic will provide complete picture of lessons and programmatic implications in improving vaccination services during pandemic situation. Overall, the study provides valuable insights for barriers in uptake immunization services during pandemic situation with high applicability for local and global context. The comments were shown below.

Thank you so much to reviewer 1 for taking the time to read and review our manuscript.

Introduction

•The problem is properly introduced and the justification is robust enough to support the objective of the study.

Methods

•Line 97: Why the study included only 18 months or under children although GP surgery provided immunization up to more than 3 years old?

• During the COVID-19 pandemic, the priority for vaccination services is to maintain delivery of vaccinations for babies, infants and pre-school children.

• The recruitment for our study specifically focused on parents and guardians with children aged 18 months or under as most childhood vaccinations delivered in general practice are due before a child reaches 18 months of age (i.e. vaccinations are scheduled at 8 weeks, 12, weeks, 16 weeks and 12 months). There is then a gap in when routine childhood vaccinations are due until a child reaches 3 years and 4 months.

• As we wanted to find out about parents and guardians views and experiences of accessing routine childhood vaccinations we chose to focus on this time period (18 months or under) as one in which children would most likely be due a vaccination. 

• The inclusion criteria were discussed with immunisation representatives at Public Health England.

We have added the following to the manuscript, in lines 97-104:

‘Our study specifically focused on parents and guardians with children aged 18 months or under as most childhood vaccinations delivered in general practice in the UK are due before a child reaches 18 months of age (i.e. vaccinations are scheduled at 8 weeks, 12 weeks, 16 weeks and 12 months). There is then a gap in when routine childhood vaccinations are due until a child reaches the age of 3 years and 4 months. Including children up to 18 months meant we could capture the views and experiences of parents and guardians whose children may have been overdue their 12 month vaccinations.’

•Line 143: What about the association between awareness of routine vaccination with the gender and education status of parents or guardians?

Thank you. These would have been interesting to examine. In designing the study, we chose to ask about location, ethnicity, employment status, household income, and number of children. An exploration of gender as an influencing factor in the study was not possible due to the limited number of men completing the survey, and we did not collect data on additional demographic factors that may have been of interest, such as education level, health literacy and religious beliefs. 

A note referring to this is now included in the discussion section (lines 487-490)

•In recruiting and collecting data for semi-structured interview, there is no information about characteristics of interviewers and saturation of data.

Lines 178-180: ‘Interviews were conducted between 27th April and 27th May 2020 by SB and PP, qualitative researchers who have extensive experience in conducting interviews on the topic of vaccination with parents.’

Through the analysis of free-text responses and interviews the authors feel that we achieved data saturation, as no new themes were emerging from open-text or interview responses.

•Line 165: Is it online based written consent?

We have added the following to lines 173-174: ‘Written informed consent was obtained from each participant. Depending on the preference of the participant, the consent form was sent and returned via email or post.’

•Line 169 & 174: What are meanings of abbreviations; SB, PP, RC and SM-J?

These are the initials of the study researchers. 

•Line 185 & 198: Please check appropriate reference for “Error! Reference Source not found”

Thank you. We have addressed these error messages – which seem to have occurred as the paper was converted from word document to a PDF when uploaded to the journal.

•Line 194 – 197: The selection criteria for choosing 61 parents among 530 with detail contact information is not clear. Only 19 out of 61 invited (31%) were interviewed which might lead to non-response bias.

Our recruitment approach is outlined in lines 166-172:

‘Respondents who had left their details were purposively contacted based on a range of characteristics, including ethnicity, household income, and geographical location. We also purposefully aimed to interview survey participants who did not provide free-text responses and respondents whose children were overdue a vaccination, or due a vaccination within 4 weeks of taking part in the survey.’

And we have added the following to the limitations section, lines 476-479:

‘A limitation of the interviews was that only 19 of 61 invited respondents took part, which may have been partly due to respondents not seeing the invitation email or not having the time to participate. It may be that those that took part in interviews had a stronger interest in vaccinations and a different view to those that did not take part.’ 

•In table 2: There are only 2 respondents whose children were overdue a vaccination. It is questionable whether the data is saturated enough in exploring the barriers of taking vaccination among such group.

Interviews were conducted as part of the study to add to the data collected through the survey. As part of the survey, rich and detailed information was obtained in open-text responses and findings from the interviews supplemented this. Through the analysis of open-text responses and interviews the authors feel that we achieved data saturation, as no new themes were emerging from open-text or interview responses. 

We have added this to the strengths section of the paper (lines 471-475):

‘Using a mixed-methods approach in this study allowed for a more complete insight into parents’ and guardians’ views and experiences of vaccinating young children during the COVID-19 pandemic in England than could have been achieved using one method only. As part of the survey, rich and detailed information was obtained in free-text responses and findings from the interviews supplemented this.’ 

Finding

•Table 5: Why medium income level and age group 6-8 months were taken as reference in logistic model?

For the income groups the data was split into quartiles and the lower and upper qualities were compared to the middle group so as to capture differences across income. Similarly, the middle age group was taken so as to better compare between those with a high number of vaccines typically scheduled (early infancy and around 12 months) and a group where vaccines are less often scheduled. 

•Line 237 – 247: The order of factors interpreted in text is better to follow the order as shown in table.

Thank you. We have changed the order of the factors interpreted in the text to reflect the order of the table.

•Line 248 – 264: It would be better to include sub-heading for these paragraphs as these paragraphs provide information not directly linked with factors associated with lack of awareness.

Thank you. This paragraph describes in more detail factors associated with lack of awareness of vaccination service continuation. We have clarified this in the text. 

•Line 268: Please check appropriate reference for “Error! Reference Source not found”

Thank you. We have addressed these error messages – which seem to have occurred as the paper was converted from word document to a PDF when uploaded to the journal.

•Line 369: Who or which organization sent confusing text-messages?

Thank you. We have now clarified this to say (lines 386-387)

‘Several interview participants also discussed receiving confusing text-messages about their appointment from their GP practice that stated their appointments had been cancelled and they should not attend the GP practice’. 

Discussion and conclusion

•Although the scope of the study is big, the discussion provides solid messages to support the conclusion. In presenting discussion, it would be better to follow the sequence of information; capability, motivation and opportunity as written in results part for better linkage.

Thank you. We have further reflected on the order of the discussion, and have kept the current flow of information as we think that it best highlights the key issues and keeps the word count down. 

Reviewer #2: This is well-organized/planned survey for the effects of pandemic on routine immunization for infants below 18 months. This is very important point for ongoing COVID-19 pandemic (potential further wave) or other pandemics in future.

Thank you so much to reviewer 2. We appreciate you taking the time to read and review our manuscript.

Authors clearly defined their aims and methods using prompt references.

There are some minor points (maybe limitations)

- Number of interviewed participants is low; only 19. Regarding enrolment number, it is difficult to evaluate this small group response.

The interviews were conducted to supplement the survey findings, which included detailed and lengthy open-text responses, and provided a more comprehensive understanding of parents’ views and experiences. We purposively invited participants with less detailed open text responses for interview. The authors feel that due to the rich information obtained in open-text survey responses and the interviews we achieved a stage where no new themes were emerging from the data.

- Your title includes terms "parents", however as usual for studies during infancy, 90%of responses came from mothers. This is not parent's view, this is mother's perspective.

Thank you for your comment. On reflection, we agree that it is a limitation that more male participants did not take part in the study; however, the study was not just open to female participants (and some male participants did take part). We therefore want to keep the proposed title and have added to the limitations section that few male parents and guardians participated. 

- Table 4-5 (coloured-component of COM-B) is busy to follow-up, maybe move to supplementary file or need to re-organize.

We have further reflected on the layout and location of the tables, and have kept the tables within the main text as we think they are best placed here. We have condensed the quotations in table 4 to reduce the size of the table and make the table easier to read. 

- I am not sure the Journal Policy (page limitations), your Results section is too long and difficult to-follow up.

Thank you. We have shortened and changed the order of the Results section to be clearer. The length of the manuscripts fits within the journal requirements.

---

## [Decision Letter · Decision Letter 1]

3 Dec 2020

Parents’ and guardians’ views and experiences of accessing routine childhood vaccinations during the coronavirus (COVID-19) pandemic: A mixed methods study in England

PONE-D-20-26785R1

Dear Dr. Bell,

We’re pleased to inform you that your manuscript has been judged scientifically suitable for publication and will be formally accepted for publication once it meets all outstanding technical requirements.

Kind regards,

Khin Thet Wai, MBBS, MPH, MA (Population & Family Planning Resear

Academic Editor

PLOS ONE

Additional Editor Comments (optional):

Reviewers' comments:

Reviewer's Responses to Questions

**Comments to the Author**

1. If the authors have adequately addressed your comments raised in a previous round of review and you feel that this manuscript is now acceptable for publication, you may indicate that here to bypass the “Comments to the Author” section, enter your conflict of interest statement in the “Confidential to Editor” section, and submit your "Accept" recommendation.

Reviewer #2: All comments have been addressed

2. Is the manuscript technically sound, and do the data support the conclusions?

Reviewer #2: Yes

3. Has the statistical analysis been performed appropriately and rigorously? 

Reviewer #2: Yes

4. Have the authors made all data underlying the findings in their manuscript fully available?

Reviewer #2: Yes

5. Is the manuscript presented in an intelligible fashion and written in standard English?

Reviewer #2: Yes

6. Review Comments to the Author

Reviewer #2: (No Response)

7. PLOS authors have the option to publish the peer review history of their article (what does this mean?). If published, this will include your full peer review and any attached files.

Reviewer #2: No

---

## [Editor Report · Acceptance letter]

16 Dec 2020

PONE-D-20-26785R1 

Parents’ and guardians’ views and experiences of accessing routine childhood vaccinations during the coronavirus (COVID-19) pandemic: A mixed methods study in England 

Dear Dr. Bell:

I'm pleased to inform you that your manuscript has been deemed suitable for publication in PLOS ONE. Congratulations! Your manuscript is now with our production department. 

Kind regards, 

on behalf of

Dr. Khin Thet Wai 

Academic Editor

PLOS ONE